# Reactivity of N-Heterocyclic Stannylenes: Oxidative Addition of Chalcogen Elements to a Chiral NH-Sn System

Kerry R. Flanagan [1], James D. Parish [1], Gabriele Kociok-Köhn [2] and Andrew L. Johnson [1,*]

1 Department of Chemistry, University of Bath, Claverton Down, Bath BA2 7AY, UK;
2 Material and Chemical Characterisation Facility (MC2), University of Bath, Claverton Down, Bath BA2 7AY, UK
* Correspondence: a.l.johnson@bath.ac.uk

**Abstract:** The reactivity of the racemic N-heterocyclic stannylene [{MeHCN($^t$Bu)}Sn] (**1**) with the chalcogenide elements $O_2$, S, Se, and Te has been investigated. In the case of the reaction of **1** with molecular oxygen, the cyclic tristannoxane complex [{MeHCN($^t$Bu)}$_2$Sn(µ-O)]$_3$ (**3**) was isolated and characterised. NMR studies ($^1$H, $^{13}$C, and $^{119}$Sn) show the formation of $D_3$- and $C_2$- symmetric assemblies. The reaction of **1** with S, Se, and Te, respectively, yielded the cyclo-distannachalcogenide complexes, [{MeHCN($^t$Bu)}$_2$Sn(µ-E)]$_3$ (**4**: E = S, **5**: E = Se, **6**: E = Te), again with multinuclear NMR studies proving the formation of $C_2$- and $C_s$-symmetric assemblies. Single crystal X-ray diffraction studies have been used to elucidate the molecular structures of the products of oxidative addition, **3**, **4**, **5**, and **6**.

**Keywords:** stannylene; chalcogen; oxidative addition

## 1. Introduction

The chemistry of heavy carbene analogues [MR$_2$:M = Si, Ge, Sn, Pb; R = stabilising anionic ligand] has attracted considerable attention for some time now and is of fundamental interest in main group p-block chemistry [1]. Their dual Lewis acidic (the presence of a vacant p-orbital) and Lewis basic characteristics (an "inert" lone pair of electrons with significant s-character), alongside steric and electronic tuneability, via judicious ligand selection, results in diverse and often unique chemistry [1–4]. While the chemistry of group 14 N-heterocyclic metallylenes has focused largely on the lighter congeners, specifically N-heterocyclic carbenes [5,6], N-heterocyclic silylenes [7–9] and N-heterocyclic germylenes [10–13], recent reports have extended to the two heaviest elements in the group and the chemistry of N-heterocyclic stannylenes [14–17] and plumbenes [18], respectively.

While N-heterocyclic metallylenes (NHMs, where M = Si, Ge, Sn, or Pb, Scheme 1) are an emerging class of compounds that have found application as versatile ligands for homogeneous catalysis [5,9], the chemistry of heavier homologues of NHCs (Ge, Sn, and Pb) are still less well understood than their carbon and silicon-based counterparts [19–23].

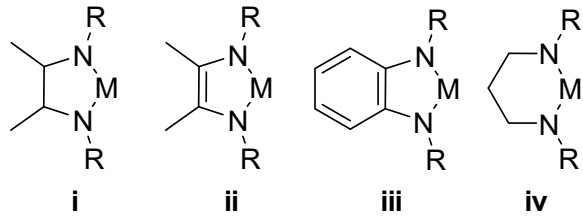

**Scheme 1.** Selected examples of non-carbon N-Heterocyclic Metallylene Systems (M = Si, Ge, Sn, or Pb; R = alkyl, aryl, or silyl group).

Of the systems shown in Scheme 1, the best-known examples are those based on the imidazol-2-ylidene framework (Scheme 1, type **i**, M = C) first reported by Arduengo et al. in 1991 [24]. The corresponding heavier analogues show a high degree of thermal stability compared to the corresponding species with unsaturated backbones (**ii**); the degree of aggregation displayed by these species is closely related to the steric bulk of the substituents on the N centers. Interest in these systems has stimulated further curiosity into related cyclic diamide systems such as the benzimidazole-like systems (**iii**) [22,25–27] and 6-membered N-heterocyclic metallylenes (**iv**) [16].

Such systems can be tuned with respect to their volatility, thermal stability, and chemical reactivity and have been found to be suitable precursors for application in thin film deposition processes such as atomic layer deposition (ALD) [18,28–32]. Accordingly, Ge, Sn, and Pb systems based around the saturated N-heterocyclic scaffolds (**i**) have been investigated as potential vapor-phase precursor compounds to produce metal chalcogenide and metal oxide thin films, $[M_xE_y]$ (M = main group metal; E = O, S, Se), e.g., SnO, SnS, $Sn_2O_3$, $Sn_2S_3$, $SnO_2$, and SnS. These materials have attracted increasing interest over recent years because of their electrical and optical properties [33–38].

However, for such potentially useful materials, relatively little is known about their chemistry. Oxidation reactions of tin(II) compounds are now an extensively studied area [1,3,4]. We have previously described the oxidative addition chemistry of N-supported stannylene complexes with elemental species (sulfur, selenium tellurium, bromine), diphenyl dichalcogenide reagents $Ph_2E_2$ (E = S, Se, and Te), as well as chalcogen transfer reagents (i.e., $SC_3H_6$ and Se=PEt$_3$) and free radical species (TEMPO) [39–41]. Such systems have potential utility in the formation of Sn(II) chalcogenide thin films and nanocrystal formation [40]. One crucial requirement for any prospective precursor compound for a process such as ALD is that it should be highly reactive not only toward substrate surfaces but also toward co-reagents such as $H_2O$, $H_2S$, $H_2O_2$, $O_2/O_3$, $NH_3$, or chalcogens and chalcogen-containing reagents (e.g., tellurium–alkylsilyl and selenium–alkylsilyl compounds) [42].

The N-heterocyclic stannylene (NH-Sn) complex, *rac*-1,3-di-tert-butyl-4,5-dimethyl-1,3-diaza-2-stannacyclopentane-2-ylidene (1), independently reported by both Gordan et al. [28,31] and Mansell et al. [15] has subsequently been shown by Gordan et al. to be a suitable ALD precursor for the deposition of SnS thin films [31]. The coordination chemistry of the NH-Sn complex **1** with [Fe(CO)$_4$] and [CpMn(CO)$_2$] fragments has also been described [15]. Here, we describe the synthesis of the new achiral N-heterocyclic stannylene, **2**, and the reaction chemistry of the known chiral N-heterocyclic stannylene, **1**, (Scheme 2), with elemental oxygen, sulfur, selenium, and tellurium.

**Scheme 2.** Synthesis of the N-heterocyclic stannylene complexes (**1**) and (**2**) (R = Me or SiMe$_3$).

## 2. Results and Discussion

The N-heterocyclic stannylenes **1** and **2** were synthesised by the reaction of the dilithium salt of the appropriate diamine with SnCl$_2$ in diethyl ether at −78 °C (Scheme 2). This produced deep red and deep yellow/orange-colored solutions, respectively. In the case of **1**, the solvent was removed in vacuo after 2 h, and the solids were extracted into n-hexane; crystallisation at −20 °C yielded dark red crystals of the pure stannylene. In the case of **2**, a hexane solution was filtered and concentrated in vacuo. Crystallisation at −28 °C yielded orange crystals of pure stannylene. In both cases, complexes **1** and **2** were also

obtained by reaction of the appropriate diamine with either *bis*[(trimethylsilyl)amido]tin(II) or *bis*(dimethylamido)tin(II) in toluene at low temperature.

The $^1$H NMR spectra (C$_6$D$_6$, 25 °C) of both **1** and **2** show only one signal for the $^t$Butyl groups (1: δ = 1.25 ppm; 2: δ = 1.36 ppm). Furthermore, the $^1$H NMR spectrum of **1** shows a doublet associated with the methyl groups of the *bis*-amide ligand (δ = 1.20 ppm, $^3$J$_{(1H-1H)}$ ≈ 6.0 Hz), as well as a single quartet associated with the signals from the {CH} groups adjacent to the nitrogen atoms (δ = 3.32 ppm, $^3$J$_{(1H-1H)}$ ≈ 6.2 Hz). The $^1$H NMR spectrum of **2** shows the presence of four separate multiplets (δ = 4.50, 3.58, 3.23, and 2.73 ppm), as expected for a $C_2$-symmetric system, indicative of a more complicated solution state structure. In addition, the $^1$H NMR spectrum for **2** shows a single broad peak (δ = 1.36 ppm} associated with the $^t$Butyl groups. The associated $^{13}$C NMR spectrum shows two broad peaks (δ = 29.6 and 32.90 ppm) representative of two distinct $^t$Butyl environments, alongside two peaks for the quaternary carbons of the tert-butyl group (δ = 47.04 and 54.05 ppm), and a broad, shallow resonance for the {CH$_2$} carbon atoms (δ = 57.99 ppm). The $^{119}$Sn NMR spectrum of **1** showed a singlet resonance at δ = 454 ppm, which is a considerably lower chemical shift compared to the stannylene [Sn{N(SiMe$_3$)$_2$}$_2$] (δ = 768 ppm), consistent with a two-coordinate tin centre. In contrast, the $^{119}$Sn NMR spectrum of **2** showed a single up-field resonance (δ = 175 ppm), indicative of an Sn(II) centre with a higher coordination number.

Use of the chiral racemic ligand N,N'-di-tert-butyl-2,3-dimethylenediamide ligand has previously been shown to have a significant effect on oligomerisation, resulting in the formation of the chiral monomeric Sn(II) complex, **1**, with both R,R and S,S $C_2$-symmetric enantiomers present in the asymmetric unit cell in a 1:1 ratio [15].

Contrastingly, the single crystal X-ray diffraction experiments carried out on **2** reveal two stannylene units linked through amide groups to form a four-membered ring complex with a puckered conformation (angle between the planes defined by N(2)–Sn(1)–N(4) and N(2)–Sn(2)–Sn(4); 25.8°). A thermal ellipsoid plot of the molecular structure of **2** is shown in Figure 1. Selected structural parameters are presented in the legend of Figure 1; experimental crystallographic details are summarised in the ESI (Table S1).

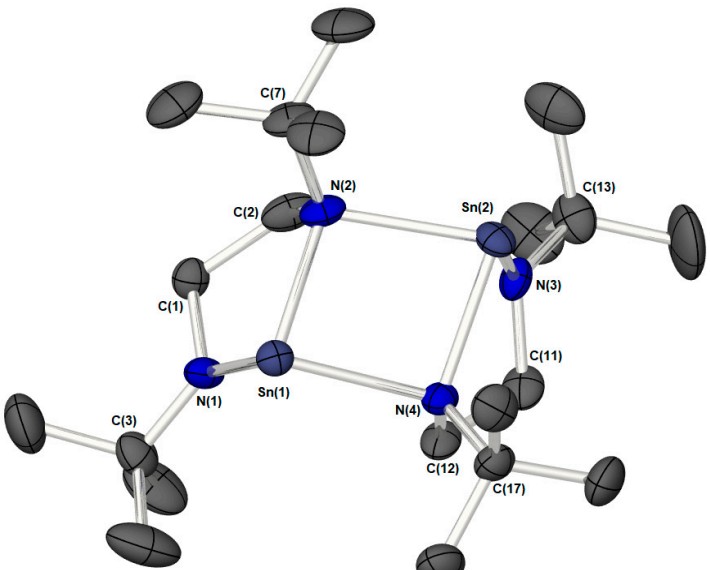

**Figure 1.** Molecular structure of **2** in the solid state (thermal ellipsoids at 50%, H atoms are omitted for clarity). Selected bond lengths [Å] and angles [°]: Sn(1)–N(1) = 2.068(3), Sn(1)–N(2) = 2.236(6), Sn(1)–N(4) = 2.323(2), Sn(2)–N(2) = 2.318(3), Sn(2)–N(3) = 2.083(2), Sn(2)–N(4) = 2.248(2), N(1)–Sn(1)–N(2) = 82.21(10)°, N(3)–Sn(2)–N(4) = 81.79(10)°, N(2)–Sn(1)–N(4) = 80.65(9)°, N(2)–Sn(2)–N(4) = 80.51(9)°, Sn(1)–N(2)–Sn(2) = 96.30(9)°, and Sn(1)–N(4)–Sn(2) = 95.84(8).

The *cis*-geometry of **2** is common to only one other known dimeric N-heterocyclic *bis*amido stannylene system, i.e., [{(CH$_2$)$_3$(NSiMe$_3$)$_2$Sn}$_2$] [43]. Common to both *cis*- and *trans*- orientated N-hetrocyclic *bis*amido stannylene systems [14,17,43,44], **2** possesses three different types of tin–nitrogen bonds. For each tin atom, the shortest of these is associated with the bond made to a nonbridging nitrogen atom: [Sn(1)–N(1) 2.088(3) Å] and [Sn(2)–N(3) 2.083(3)], respectively. The intermediate Sn–N bond is made within a monomeric subunit to the other bridging nitrogen atom [Sn(1)–N(2) 2.236(6) Å and Sn(2)–N(4) 2.248(2) Å]. The longest Sn–N bonds are formed between two subunits to generate the dimeric structure [Sn(1)–N(4) 2.323(2) Å; Sn(2)–N(2) 2.318(3) Å]. These bond lengths are in contrast to the shorter Sn–N bonds observed in **1** [Sn–N$_{ave}$ ~2.032 Å] [15,28].

Within the five-membered {SnN$_2$C$_2$}-rings, the internal N–Sn–N bite-angles of the diamide ligand, N(1)–Sn(1)–N(2) and N(3)–Sn(2)–N(4) are 82.21(9)° and 81.79(10)°, respectively, and are comparable to corresponding angles in the monomeric NH-Sn system **1** (*cf.* ~82) [15,28,31].

Whilst the three coordinate N-atoms are approaching planarity [$\Sigma_{N(1)}$ = 355.7(2)° and $\Sigma_{N(3)}$ = 350.95(19)°], the four coordinate nitrogen atoms are significantly more pyramidalised [N(2): $\tau_4$ = 0.93; N(4): $\tau_4$ = 0.93] [45]. In comparison, the N atoms observed in **1** are, within experimental error, planar [$\Sigma_{N(ave)}$ = 359.8°]. The coordinative differences between complexes **1** and **2** are clearly due to the steric effect of the methyl groups vs H atoms on the two C$_2$-bridge backbones, i.e., the presence of the methyl groups in **1** force planarity at the nitrogen atoms prohibiting pyramidalisation and subsequent dimer formation, *cf* complex **2**.

Despite NH-E systems of group 14 elements having been known and reported on since 1975 [46], one of the most obvious transformations, namely their oxidation with chalcogens, was not investigated until a decade later, when Veith et al. reported the reactions of a germanium(II) *bis*(tert-butylamido)cyclodisilazane with oxygen and sulfur [47]. Since then, a number of NH-E systems, including *bis*(amido) complexes, [M{HMDS}$_2$] (M = Ge, Sn; HMDS = N(SiMe$_3$)$_2$), have also been shown to undergo reaction with heavier chalcogenide elements (E = S, Se, Te) to yield the bridged dimers [($\mu^2$-E)M{HMDS}$_2$]$_2$ [41,48]. However, the reaction of Sn(II) species with elemental chalcogens is typically unpredictable and can result in a range of systems, including both bridging and terminal {E$^{2-}$} fragments as well as chelating {E$_2{}^{2-}$} and {E$_4{}^{2-}$} groups [39,49]. Although transition metal complexes containing chalcogen and poly-chalcogen ligands are numerous, there is a general paucity of examples containing main group metals [41,49–53].

Direct reaction of O$_2$ with divalent Sn(II) systems is even less common, with only a handful of systems reported [40,54,55]. Bubbling of O$_2$ through a hexane solution of **1** results in an immediate colour change from red/orange to pale yellow, Scheme 3. After 2 h, the solvent was removed in vacuo, and the solids were extracted into fresh n-hexane and filtered. Crystallisation at −28 °C yielded pale yellow crystals of a new tin complex **3**.

**Scheme 3.** Synthesis of the cyclo-tristannoxane complex **3**, upon the reaction of a racemic mixture of the chiral NH-stannylene **1** with molecular oxygen.

Given the racemic yet chiral nature of **1**, the oxidative product **3** displays a more complicated $^1$H NMR spectrum (in $C_6D_6$) with the presence of a multiplet ($\delta$ = 1.13–1.18 ppm, 6H) associated with the methyl groups of the *bis*-amide ligand, alongside two sharp singlets ($\delta$ = 1.35 and 1.36 ppm, 18H in a 10:8 ratio) and a complex multiplet associated with the {CH} units of the amide ligand backbone ($\delta$ = 2.81–2.91 ppm). The $^{13}$C NMR spectrum of **3** clearly shows the presence of more than one species in the solution state, with four peaks associated with the methyl groups of the *bis*-amide backbone ($\delta$ = 26.83, 26.94, 26.96, and 27.00 ppm), four peaks associated with the tert-butyl groups (both {CH$_3$} and {C}) ($\delta$ = 33.40, 33.50, 33.7, 33.75 ppm and $\delta$ = 53.37, 53.45, 53.53, and 55.55 ppm, respectively) and four peaks associated with the methine carbon in the *bis*-amide backbone ($\delta$ = 55.71, 55.82, 55.91, and 56.37 ppm).

Similarly, the $^{119}$Sn NMR spectrum shows the presence of three distinct single up-field resonance peaks ($\delta$ = −188, −189, and −191 ppm), with the latter of these resonances displaying a higher peak intensity. Single crystal X-ray diffraction experiments carried out on **3** reveal the molecular structure to be a rare example of an acyclo-tristannoxane system, formed from direct oxidation of a stannylene, consisting of a planar six-membered [Sn$_3$O$_3$] core and crystalising in the triclinic space group P-1. Each tin(IV) center is four coordinates, attached to both the chelating [*rac*-(Me)$_2$H$_2$C$_2$(N$^t$Bu)$_2$], ligand, and two oxygen atoms, as seen in Figure 2. A thermal ellipsoid plot of the molecular structure of **3** is shown in Figure 2, and selected structural parameters are presented in the legend of each figure; crystallographic data are provided in Table S1 (ESI).

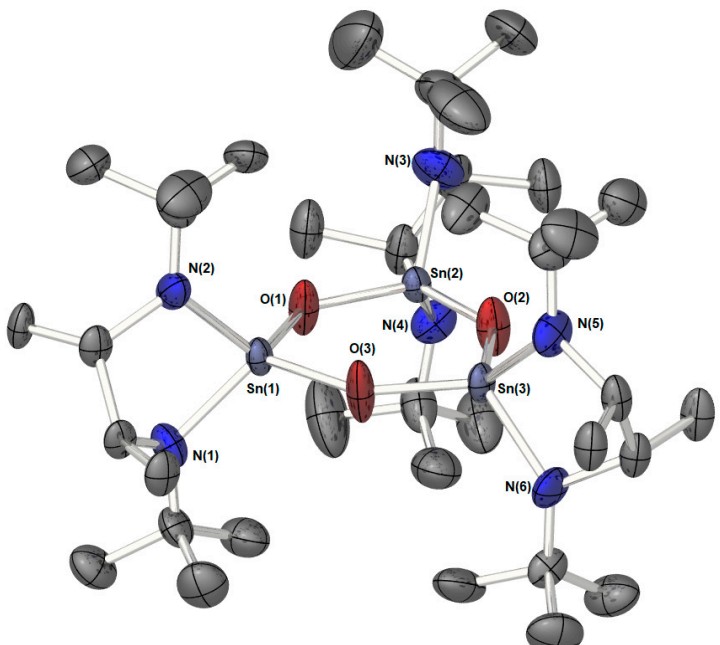

**Figure 2.** Molecular structure of the $R_6$ cyclo-tristannoxane complex [{μ-OSn(κ$^2$-(NtBu)$_2$C$_2$H$_2$Me$_2$)}$_3$] $R_6$-**3** in the solid state. Thermal ellipsoids are shown at 50%. H atoms and disorder in the {(Me)HCCH(Me)} backbone are omitted for clarity. Selected bond lengths [Å] and angles [°]: Sn(1)–O(1) 1.936(4), Sn(2)–O(1) 1.949(4), Sn(2)–O(2) 1.934(4), Sn(3)–O(2) 1.950(4), Sn(3)–O(3) 1.926(5), Sn(1)–O(3) 1.951(5), Sn(1)–N(1) 2.009(5), Sn(1)–N(2) 2.002(5), Sn(2)–N(3) 1.995(6), Sn(2)–N(4) 2.004(5), Sn(3)–N(5) 2.009(5), Sn(3)–N(6) 1.998(6); Sn(1)–O(1)–Sn(2) 135.41(19), Sn(2)–O(2)–Sn(3) 135.59(19), Sn(3)–O(3)–Sn(1) 135.79(19), O(1)–Sn(1)–O(3) 104.32(19), O(1)–Sn(2)–O(2) 104.41(19), O(2)–Sn(3)–O(3) 104.30(19), N(1)–Sn(1)–N(2) 85.90(19), N(3)–Sn(2)–N(4) 85.31(19), N(5)–Sn(3)–N(6) 86.20(19).

The Sn–O bond lengths, which are within the range 1.936(4)–1.949(4) Å, are slightly shorter than for related tin(IV) complexes; e.g., (Sn–O) 1.987(7) Å for [(μ-O$_2$)Sn{N(SiMe$_3$)$_2$}$_2$,]$_2$ [40] or 1.96–1.99 Å for the cyclo-tristannoxane [(μ-O)(SnR$_2$)]$_3$ (R = $^t$Bu, R= 2,6-C$_6$H$_3$Et$_2$ or R$_2$ = {(NSiMe$_3$)$_2$C$_{20}$H$_{12}$}) [56–58]. On average, the O-Sn-O$_{ave}$ angles [104.3°] are marginally

smaller than an ideal tetrahedral value. Contrastingly, the N–Sn–N$_{ave}$ angles are considerably smaller [85.8°]. The Sn–O–Sn angles range from 135.41(19)° to 135.79(19)°, rendering the endocyclic angles at tin narrower than those at oxygen. The Sn–N bond lengths of **3** are between 1.998(6) and 2.009(5) Å, and are expectedly shorter than Sn–N bonds observed in the amidostannylenes, [Sn{N(SiMe$_3$)$_2$}$_2$] or **1**, (2.096(1)/2.088(6) Å, and 2.038(4)/2.033(5) Å, respectively) and are more comparable to the Sn–N bonds observed for the Sn(IV) amide oxo/per-oxo/alkoxide systems [(µ-O$_2$)Sn{HMDS}$_2$]$_2$ ((Ave.) 2.013(2) Å), [(TEMPO)$_2$Sn{HMDS}$_2$] ((Ave.) 2.0553(15) Å), and [(Me$_2$N)$_2$Sn(OC$_6$$^t$Bu$_2$MeH$_2$)$_2$] ((Ave.) 1.998(4) Å) [39,59].

The presence of an equal mix of *SS* and *RR* diastereoisomers of **1** in the reaction with dioxygen results in the assembly of chiral cyclo-tristannoxane complexes. Oligomerisation of three "homochiral" stannoxane units [O=Sn(κ$^2$-(N$^t$Bu)$_2$C$_2$H$_2$Me$_2$))] (either *RR* or *SS* chirality) affords the *D$_3$*-symetric assemblies, *S$_6$*-**3** and *R$_6$*-**3**, as shown in Scheme 4.

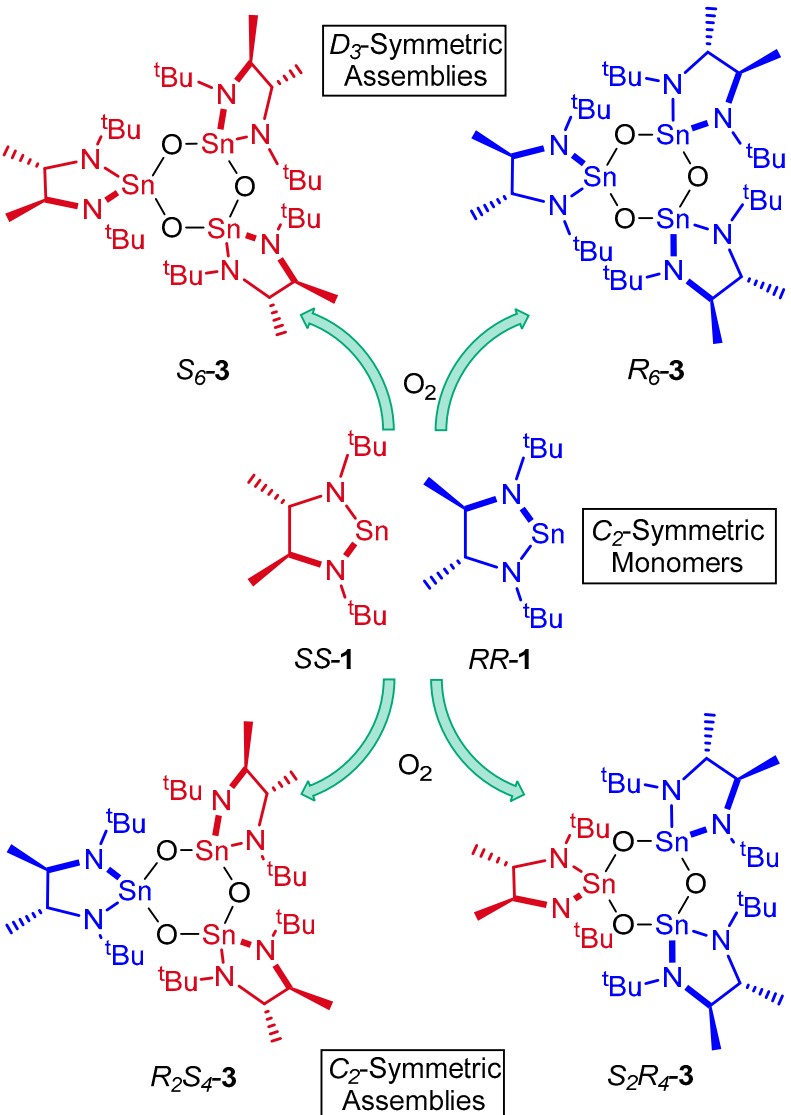

**Scheme 4.** Chirality-generated cyclo-tristannoxane assemblies (**3**) formed a racemic mixture of *SS*-**1** and *RR*-**1** upon reaction with molecular oxygen.

In the same way, chiral arrangements with C$_2$-symmetry are generated by "heterochiral" coordination, where the one stannoxane unit in the cyclic trimer of opposite chirality to the other two units such that cyclic systems with *R$_2$S$_4$*-**3** and *S$_2$R$_4$*-**3** symmetry

are formed (Scheme 4). Similar dynamic assemblies have been observed in macrocyclic chemistry and result in a statistical distribution of chiral forms [60]. The presence of both the helical $D_3$-symetric $S_6$-**3** and $R_6$-**3** molecules, alongside the $C_2$-symetric $R_2S_4$-**3** and $S_2R_4$-**3** in the solid state, manifests as disorder in the *bis*-amide backbone {(Me)HCCH(Me)}. However, the presence of these various chiral forms is clearly observed in multinuclear NMR studies, i.e., the presence of three distinct resonances in the $^{119}$Sn NMR is consistent with the presence of both the $S_6$-**3** and $R_6$-**3** forms, which have identical chemical shifts [$\delta = -190.6$ ppm], alongside the chiral cyclic tristannoxane forms $R_2S_4$-**3** and $S_2R_4$-**3** [$\delta = -188.3$ and $-189.1$ ppm].

Given these observations, attempts were made to further investigate **3** by VT-$^1$H NMR. Raising the temperature of the sample (353 K in $C_6D_6$) had little effect on the spectra, with only a slight shifting and broadening of the peaks at 2.83–2.90 ppm associated with the {CH} of the *bis*-amide backbone. Lowering the temperature of the sample from 296 K to 273 K (in $d_8$-Tol) had no discernable effect on the NMR spectra, and attempts at lower temperature studies (~233 K) were limited by solubility issues.

Our observation that complex **1** reacts with $O_2$ to form the cyclo-tristannoxane assembly, **3**, is consistent with previously reported DFT calculations which speculate upon the possible mechanism for the atomic layer deposition of $SnO_x$ from the reaction of the related N-heterocyclic N,N'-$^t$butyl-1,1-dimethylethylenediamine stannylene (**I**) (Scheme 5) with ozone [61]. DFT calculations by Park et al. suggests that the reaction of the stannylene (**I**) to form $SnO_2$ proceeds via oxidation of the Sn(II) center and formation of Sn(IV) products.

**Scheme 5.** DFT-predicted reactivity of the stannylene (**I**) with $O_3$ [61].

Direct reaction of the NH-stannylene **1** with two equivalents of the elemental chalcogenides S, Se, or Te, respectively, results in the formation of the bimetallic Sn(IV) chalcogenide complexes [{μ-E-Sn($\kappa^2$-(N$^t$Bu)$_2$C$_2$H$_2$Me$_2$)}$_3$] (**4**: E = S, **5**: E = Se, **6**: E = Te) as yellow to red crystalline compounds, Scheme 6. In the case of sulfur, the reaction between **1** and an excess S (2, 4, and 6 equivalents, respectively) was also investigated in an attempt to synthesise larger poly-chalcogen-containing species. In each case, the reaction resulted in the formation and isolation of the mono-sulfide complex **4** alongside $S_8$, as identified by single-crystal X-ray diffraction.

**Scheme 6.** Synthesis of the cyclo-distannachalcogenide complexes, **4–6**, upon reaction of chiral NH-stannylene **1** with stoichiometric (1:1) amounts of elemental S (**4**), Se (**5**), and Te (**6**), respectively.

Multinuclear NMR data for compounds **1**, **4–6** are summarised in Table 1. $^1$H and $^{13}$C NMR spectra of the products **4–6** showed resonances consistent with the observed product. As with complex **3**, the $^1$H NMR spectra for **4–6** are comparable to that of the starting material, **1**, with only one set of resonances for the {Me}, {$^t$Bu}, and {CH} functionalities observed. Close inspection of the $^1$H NMR spectra shows $^3J_{1H-119Sn}$ coupling (approx. ~105 Hz (**4**), ~95 Hz (**5**), and 85 Hz (**6**)) between the methine {CH} group, in the *bis*-amide backbone, and the central Sn(IV) center to which it is coordinated.

**Table 1.** $^1$H, $^{13}$C, and $^{119}$Sn NMR chemical shifts for the tin compounds **1**, **4**, **5**, and **6**.

| Compound [a] | $\delta(^1$H$)$ [b] | $\delta(^{13}$C$)$ [c] | $\delta(^{119}$Sn$)$ [d] |
|---|---|---|---|
| **1** | 1.20 (d, 6H, $^3J_{H-H}$ = 6.2 Hz, {C*Me*H})<br>1.25 (s, 18H, {N-C*Me*$_3$})<br>3.32 (q, 2H, $^3J_{H-H}$ = 6.1 Hz, {CMe*H*}) | 29.0 ({C*Me*H})<br>34.5 ({N-C*Me*$_3$})<br>56.3 ({*C*MeH})<br>64.8 ({N-*C*Me$_3$}) | 455 |
| **4** | 1.11 (d, 6H, $^3J_{H-H}$ = 5 Hz, {C*Me*H})<br>1.47 (s, 18H, {N-C*Me*$_3$})<br>2.74–3.02 (m, 2H, {CMe*H*}) | 26.5 ({C*Me*H})<br>26.6 ({C*Me*H})<br>33.3 ({N-C*Me*$_3$})<br>33.4 ({N-C*Me*$_3$})<br>55.4 ({N-*C*Me$_3$})<br>55.5 ({N-*C*Me$_3$})<br>58.3 ({*C*MeH})<br>54.4 ({*C*MeH}) | −79 (J$_{119Sn-117Sn}$ = 589 Hz)<br>−84 (J$_{119Sn-117Sn}$ = 582 Hz) |
| **5** | 1.11 (d, 6H, $^3J_{H-H}$ = 5 Hz, {C*Me*H})<br>1.47 (s, 18H, {N-C*Me*$_3$})<br>2.82–2.86 (m, 2H, {CMe*H*}) | 26.2 ({C*Me*H})<br>26.7 ({C*Me*H})<br>33.3 ({N-C*Me*$_3$})<br>33.4 ({N-C*Me*$_3$})<br>55.8 ({N-*C*Me$_3$})<br>55.9 ({N-*C*Me$_3$})<br>58.6 ({*C*MeH})<br>58.7 ({*C*MeH}) | −357<br>−362 |
| **6** | 1.08 (d, 12H, $^3J_{H-H}$ = 5 Hz, {C*Me*H})<br>1.55 (s, 32H, {N-C*Me*$_3$})<br>2.77 (q, 4H, $^3J_{H-H}$ = 5 Hz, {CMe*H*}) | 26.7 ({C*Me*H})<br>26.8 ({C*Me*H})<br>33.5 ({N-C*Me*$_3$})<br>33.7 ({N-C*Me*$_3$})<br>56.5 ({N-*C*Me$_3$})<br>59.0 ({*C*MeH})<br>59.1 ({*C*MeH}) | −957<br>−961 |

[a] All NMR spectra were obtained in C$_6$D$_6$ at 296 K. [b] 500 MHz, [c] 125.7 MHz, [d] 186.5 MHz.

A more accurate resolution is frustrated by the presence of $C_2$-symmetric homochiral ($R_4/S_4$) and $C_S$-symmetric heterochiral ($R_2S_2/S_2R_2$), which overlap in the spectrum (Scheme 7). In contrast, both the $^{13}$C NMR and $^{119}$Sn spectra display multiple resonances indicative of the presence of the $C_2$-symmetric homochiral ($R_4/S_4$) and $C_S$-symmetric heterochiral ($R_2S_2/S_2R_2$ which are superimposable and therefore equivalent) species (Scheme 7).

The values of $\delta(^{119}$Sn$)$ correlate with the electronegativity of the chalcogenide (**4**: −79/−84, **5**: −357/−362, and **6**: −957/−961) with the small difference in chemical shift resulting from differing molecular conformations, and are comparable to related Sn(IV) *bis*-amide chalcogenide dimers [41].

Uniquely in this study, the $^{119}$Sn NMR spectra of the sulfide complex **4** clearly shows $^{119}$Sn-$^{117}$Sn coupling (See Table 1). For complexes **5** and **6**, it was not possible to detect either $^{119}$Sn–$^{77}$Se or $^{119}$Sn–$^{125}$Te coupling. As with complex **3**, attempts to interrogate complexes **4–6** using VT-$^1$H NMR were unsuccessful: high-temperature $^1$H NMR studies (343 K) (C$_6$D$_6$) showed no significant changes to the spectra. Low temperature (<273 K) studies (in d$_8$-Tol) were thwarted by solubility issues. Unfortunately, an exhaustive investigation of compounds **5** and **6** by $^{77}$Se and $^{125}$Te NMR spectroscopy, respectively, failed to reveal

the anticipated Se or Te resonances, which are commonly found over a very large chemical shift range.

**Scheme 7.** Chirality-generated cyclo-distannachalcogenides complex, **4–6,** formed a racemic mixture of *SS*-**1** and *RR*-**1** upon reaction with chalcogenide elements (NB: $R_2S_2 \equiv S_2R_2$).

Figure 3 shows the molecular structure of the $S_2R_2$-**5** isomer (as a representative example of complexes **4–6**), selected structural parameters for **4–6** are presented in the legend of the figure. Single crystal X-ray diffraction experiments were carried out on **4–6** and revealed these complexes to be isostructural examples of cyclo-distannachalcogenides, consisting of a central, planar, four-membered $[Sn_2E_2]$ core (**4**: E = S, **5**: E = Se, **6**: E = Te), crystalising in the monoclinic space group $P2_1/n$, such that the complexes possess a crystallographic inversion centre at the midpoint of the central $[Sn_2E_2]$ ring. As with complex **3**, the racemic nature of complexes **4**, **5**, and **6**, i.e., the presence of equal amounts of the $R_4$, $S_4$, $R_2S_2$, and $S_2R_2$ geometric assemblies (Scheme 7) results in crystals that crystalise in a racemic space-group, with the associated appearance of disorder in the *bis*-amide ligand.

The bulky *bis*-amide ligands are attached to each 4-coordinate Sn atom such that the {N–Sn–N} plane of the NH–Sn ring is approximately perpendicular to the $[Sn_2E_2]$ core [**4**: 87.9(5)°, **5**: 88.9(3)°, **6**: 88.4(6)°].

The Sn-E bond lengths increase with atomic number of E and are within the range expected for single bonds: Sn–$S_{ave}$ (~2.4256(5) Å), Sn–$Se_{ave}$ (~2.5483(3) Å), and Sn–$Te_{ave}$ (~2.7527(3) Å) and are comparable to structurally related systems [39–41]. The Sn–$N_{ave}$ bond lengths for compounds **4–6** are reduced by ca. 0.01–0.02 Å, and the N–Sn–$N_{ave}$ angles are increased by ca. 3.7–5.2°, compared with the values for the tin(II) starting material, **1**, [Sn–$N_{ave}$ = 2.0323(4) Å and N–Sn–$N_{ave}$ = 82.01(17)°]. This variation is attributed mainly to the radius of $Sn^{4+}$ being smaller than that of $Sn^{2+}$, but a steric effect may also be a contributory factor. Concomitantly, the endocyclic E–Sn–E bonds become more acute as the atomic number of E increases (**4**: 86.832(19)°, **5**: 84.773(9)°, **6**: 82.163(8)°). In each of the three compounds, **4–5**, the N atoms have an approximately trigonal planar environment. It is worth noting that the molecular structure of the related Ge(II) analogue of (6) has been reported as a private communication to the Cambridge structural database, i.e., CSD-ATEBAF (see crystallography section).

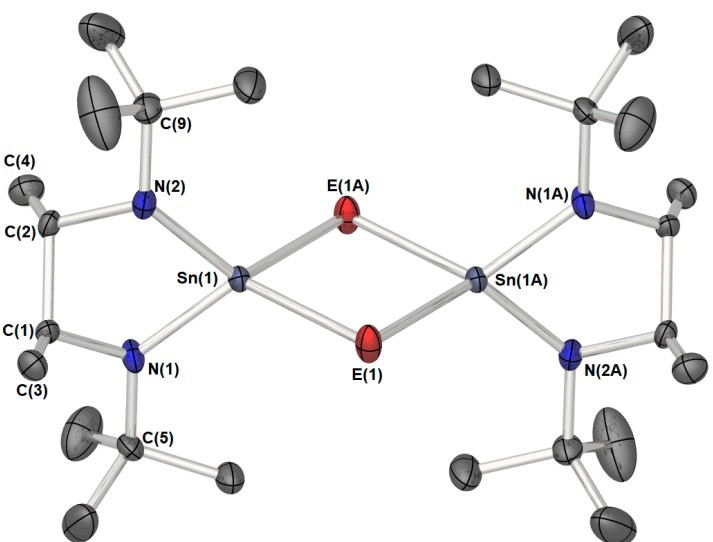

**Figure 3.** Molecular structures of the selenium derivative complex **5** ($S_2R_2$-isomer). Hydrogen atoms and disorder in the {(Me)HCCH(Me)} backbone have been omitted for clarity. Thermal ellipsoids are shown at 50%. Symmetry transformations used to generate equivalent atoms: (A) $1 - x$, $-y$, $1 - z$. Selected bond lengths [Å] and angles [°] **4**: Sn(1)–S(1) 2.4240(5), Sn(1)–S(1A) 2.4272(5), Sn(1)–N(1) 2.010(3), Sn(1)–N(2) 2.013(3), Sn(1)···Sn(1A) 3.334(3); S(1)–Sn(1)–S(1A) 93.167(17), N(1)–Sn(1)–N(2) 86.4(4) Sn(1)–S(1)–Sn(1A) 86.832(17); **5**: Sn(1)–Se(1) 2.5448(3), Sn(1)–Se(1A) 2.5517(3), Sn(1)–N(1) 2.011(3), Sn(1)–N(2) 2.013(2) Sn(1)···Sn(1A) 3.436(3); Se(1)–Sn(1)–Se(1A) 95.227(9), N(1)–Sn(1)–N(2) 87.2(8), Sn(1)–Se(1)–Sn(1A) 84.773(9); **6**: Sn(1)–Te(1) 2.7593(3), Sn(1)–Te(1A) 2.7461(3), Sn(1)–N(1) 2.012(4), Sn(1)–N(2) 2.013(4), Sn(1)···Sn(1A) 3.618(3); Te(1)–Sn(1)–Te(1A) 97.837(8), N(1)–Sn(1)–N(2) 85.7(6), Sn(1)–Te(1)–Sn(1A) 82.163(8).

Despite the fact that reactions of the methyl-substituted *bis*-amide complex **1** were successful, attempts to react the less sterically encumbered stannylene, complex **2**, with stoichiometric amounts of elemental O, S, Se, or Te, respectively, were all unsuccessful, yielding intractable materials. We attribute this difference to the presence of the {$CH_2CH_2$} in the backbone of **2,** which presumably affects both solubility of products and the degree to which the products aggregate and oligomerise.

## 3. Materials and Methods

**Experimental Details**. Elemental analyses were performed using an Exeter Analytical CE 440 analyser. $^{1}H$, $^{13}C$ $^{119}Sn$, $^{77}Se$, and $^{125}Te$ NMR spectra were recorded on a 500 MHz Agilent ProPulse FT-NMR spectrometer, as saturated solutions at room temperature, unless stated otherwise; chemical shifts are in ppm with respect to Me$_4$Si ($^{1}H$, $^{13}C$). All reactions were carried out under an inert atmosphere using standard Schlenk techniques. Solvents were dried and degassed under an argon atmosphere over activated alumina columns using an Innovative Technology solvent purification system (SPS). The Sn(II) amides [Sn(NMe$_2$)$_2$]$_2$ [62] and [Sn{N(SiMe$_3$)$_2$}] [63] were prepared by literature methods, as was N,N′-Di-tert-butylbutylene-2,3-diamine [64]. The reagents SnCl$_2$, N,N′-Di-tert-butylethylenediamine, S$_8$, Se, and Te were purchased from Aldrich chemicals.

**[{MeHCN($^{t}$Bu)}$_2$Sn] (1)**: Under inert conditions, at $-75$ °C, *bis*[*bis*(trimethylsilyl) amido]tin(II) (4.39 g, 10 mmol) was added to a solution of {HN($^{t}$Bu)CHMe}$_2$ (2 g, 10 mmol) in Et$_2$O (60 mL), the orange solution was allowed to warm to room temperature and stir overnight. The solvent was removed under reduced pressure, and the residue was extracted with n-hexane (60 mL) and filtered through Celite. The solution was reduced in volume in vacuo and then placed at $-28$ °C overnight, upon which yellow crystals were observed (2.76 g, 8.7 mmol, 87 % yield). The product could be further purified by sublimation at $\sim 5 \times 10^{-1}$ Torr and 100 °C. Elemental Analysis for C$_{12}$H$_{26}$N$_2$Sn: Calculated (%): C 45.46,

H 8.27, N 8.84. Found (%): C 45.33, H 8.77, N 8.50. $^1$H NMR (500MHz, $C_6D_6$): δ 1.20 (d, 6H, J = 6.0 Hz, C-$\underline{Me}$), 1.25 (s, 18H, N-C$\underline{Me}_3$), 3.32 (q, 2H, J = 6.1 Hz, C-$\underline{H}$). $^{13}$C{$^1$H} NMR (125.7MHz, $C_6D_6$): δ 29.0, (s, C$\underline{Me}$), 34.5 (s, C$\underline{Me}_3$), 34.8 (s, N-$\underline{C}$Me$_3$), 56.3 (s, N-CH). $^{119}$Sn{$^1$H} NMR (186.5 MHz, $C_6D_6$): δ 455.

**[{H$_2$CN($^t$Bu)}$_2$Sn]$_2$ (2):** Under inert conditions, at −75 °C, *bis*[*bis*(trimethylsilyl)amido]tin(II) (4.4 g, 10 mmol) was added to a solution of [{HN($^t$Bu)CH$_2$}$_2$] (1.72 g, 10 mmol) in Et$_2$O (60 mL), the yellow solution was allowed to warm to room temperature and stir overnight. The solvent was removed under reduced pressure, and the residue was extracted with n-hexane (60 mL) and filtered through Celite. The solution was reduced in volume in vacuo and then placed at −28 °C overnight. The yellow crystals which were formed were isolated by filtration and washed with cold hexane (−78 °C) (2.44 g, 8.4 mmol, 84 % yield). Elemental Analysis for $C_{20}H_{44}N_4Sn_2$: Calculated (%): C 41.56, H 7.67, N 9.69. Found (%): C 41.59, H 7.75, N 9.70. $^1$H NMR (500 MHz, $C_6D_6$): δ 1.36 (s, 36H, C$\underline{Me}_3$), 2.73 (br m, 2H, C$\underline{H}$), 3.23 (br m, 2H, C$\underline{H}$) 3.58 (br m, 2H, C$\underline{H}$), 4.50 (br m, 2H, C$\underline{H}$). $^{13}$C{$^1$H} NMR (125.7 MHz, $C_6D_6$): δ 29.6 (s, C$\underline{Me}_3$), 32.90 (s, C$\underline{Me}_3$) 47.04 (s, $\underline{C}$Me$_3$), 54.05 (s, $\underline{C}$Me$_3$) 57.99 (br-s, CH$_2$). $^{119}$Sn{$^1$H} NMR (186.5 MHz, $C_6D_6$): δ 175.

**[{MeHCN($^t$Bu)}$_2$Sn(μ-O)]$_3$ (3):** Excess O$_2$ gas was bubbled through an orange solution of [{MeHCN($^t$Bu)}$_2$Sn] (**1**) (0.63 g, 2 mmol) in hexane (20 mL) at 0 °C for 30 min. The resulting cloudy yellow solution was filtered, and the volume was reduced in vacuo. Storage at −28 °C over 4 days afforded pale yellow crystals (0.45 g, 72 % based on Sn) which were isolated by filtration and washed with cold (−78 °C) hexane. Elemental Analysis for $C_{36}H_{78}N_6O_3Sn_3$: Calculated (%): C 43.27, H 7.87, N 8.41. Found (%): C 43.29, H 7.85, N 8.41. $^1$H NMR (500 MHz, $C_6D_6$); δ 1.13–1.18 (m, 18H, C-$\underline{Me}$) 1.35–136, (s, 54H, C$\underline{Me}_3$) 2.90–2.83 (m, 6H, C-H). $^{13}$C{$^1$H} NMR (127.5 MHz, $C_6D_6$); δ 26.83, 26.94, 26.96, 26.99, (s, C$\underline{Me}$), 33.39, 33.50, 33.69, 33.74 (s, C-C$\underline{Me}_3$), 53.38, 53.46, 53.53, 53.55, (s, C-$\underline{C}$Me$_3$), 55.71, 55.82, 55.91, 56.37 $^{119}$Sn{$^1$H} (186.5 MHz, $C_6D_6$); δ −188, −189, −191.

**[{MeHCN($^t$Bu)}$_2$Sn(μ-S)]$_2$ (4):** A solution of [{MeHCN($^t$Bu)}$_2$Sn] (**1**) (0.63 g, 2 mmol) and elemental sulfur (0.067 g, 2 mmol) in THF (20 mL) was heated to 60 °C and sonicated for 1 hr. After stirring for 2 hr, the pale-yellow solution was filtered, and yellow crystals were obtained after crystallisation at −28 °C (0.49 g, 70 % based on Sn). Elemental Analysis for $C_{24}H_{52}N_4S_2Sn_2$: Calculated (%): C 41.28, H 7.51, N 8.02. Found (%): C 41.30, H 7.55, N 7.90. $^1$H NMR (500 MHz, $C_6D_6$); δ 1.11 (d, 6H, $^3$J$_{H-H}$ = 5 Hz, {C$\underline{Me}$H}), 1.47 (s, 18H, {N-C$\underline{Me}_3$}), 2.74–3.02 (m, 2H, {CMe$\underline{H}$}); $^{13}$C{$^1$H} NMR (127.5 MHz, $C_6D_6$); δ 26.5 ({C$\underline{Me}$H}), 26.6 ({C$\underline{Me}$H}), 33.3 ({N-C$\underline{Me}_3$}), 33.4 ({N-C$\underline{Me}_3$}), 55.4 ({N-$\underline{C}$Me$_3$}), 55.5 ({N-$\underline{C}$Me$_3$}), 58.3 ({$\underline{C}$MeH}), 54.4 ({$\underline{C}$MeH}); $^{119}$Sn{$^1$H} (186.5 MHz, $C_6D_6$); δ −79, −84.

**[{MeHCN($^t$Bu)}$_2$Sn(μ-Se)]$_2$ (5):** A solution of [{MeHCN($^t$Bu)}$_2$Sn] (**1**) (0.32 g, 1 mmol) and elemental selenium (0.08 g, 1 mmol) in THF (15 mL) was sonicated for 1 h and stirred for a further 24 hr. The apparent green solution was filtered, and pale orange crystals were obtained after crystallisation at −28 °C (0.37 g, 94% based on Sn). Elemental Analysis for $C_{24}H_{52}N_4Se_2Sn_2$: Calculated (%): C 36.39, H 6.62, N 7.07. Found (%): 36.2; H, 6.7; N, 7.5; $^1$H NMR (500 MHz, $C_6D_6$); δ 1.11 (d, 6H, $^3$J$_{H-H}$ = 5 Hz, {C$\underline{Me}$H}), 1.47 (s, 18H, {N-C$\underline{Me}_3$}), 2.82–2.86 (m, 2H, {CMe$\underline{H}$}); $^{13}$C{$^1$H} NMR (127.5 MHz, $C_6D_6$); δ 26.2 ({C$\underline{Me}$H}), 26.7 ({C$\underline{Me}$H}), 33.3 ({N-C$\underline{Me}_3$}), 33.4 ({N-C$\underline{Me}_3$}), 55.8 ({N-$\underline{C}$Me$_3$}), 55.9 ({N-$\underline{C}$Me$_3$}), 58.6 ({$\underline{C}$MeH}), 58.7 ({$\underline{C}$MeH}), $^{119}$Sn{$^1$H} (186.5 MHz, $C_6D_6$); δ −357, −362.

**[{MeHCN($^t$Bu)}$_2$Sn(μ-Te)]$_2$ (6):** Compound **6** was prepared as above using elemental tellurium (0.13 g, 1 mmol) to produce dark red crystals after crystallisation at −28 °C (0.4 g, 90 % based on Sn). Analysis: Found: C, 30.5; H, 5.8; N, 5.9. Elemental Analysis for $C_{24}H_{52}N_4Te_2Sn_2$: C, 32.4; H, 5.9; N, 6.3%. 1H NMR (500 MHz, $C_6D_6$); δ 1.08 (d, 12H, $^3$J$_{H-H}$ = 5 Hz, {C$\underline{Me}$H}), 1.55 (s, 32H, {N-C$\underline{Me}_3$}), 2.77 (q, 4H, $^3$J$_{H-H}$ = 5 Hz, {CMe$\underline{H}$}); 26.7 ({C$\underline{Me}$H}), 26.8 ({C$\underline{Me}$H}), 33.5 ({N-C$\underline{Me}_3$}), 33.7 ({N-C$\underline{Me}_3$}), 56.5 ({N-$\underline{C}$Me$_3$}), 59.0 ({$\underline{C}$MeH}), 59.1 ({$\underline{C}$MeH}); $^{119}$Sn{$^1$H} (186.5 MHz, $C_6D_6$); δ −956, −960.

**Crystallography:** Experimental details relating to the single-crystal X-ray crystallographic studies are summarised in Table S1 (ESI). Crystallographic data were collected at 150 K on a Nonius Kappa-CCD Diffractometer [λ(Mo-Kα) = 0.71073 Å], and solved by

direct methods (SIR-92) [65] and refined against all F2 using SHELXL-97 [66]. All hydrogen atoms are included in idealised positions and refined using the riding model. The structure solution was followed by full-matrix least-squares refinement and was performed using the WinGX-1.70 suite of programmes. All non-hydrogen atoms were refined anisotropically. CCDC 2259200 (**2**), 2259201 (**3**), 2259202 (**4**), 2259203 (**5**), and 2259204 (**6**) contain supplementary crystallographic data for this paper. These data, as well as the data for CSD-ATEBAF (CSD 2084275) can be obtained free of charge from The Cambridge Crystallographic Data Centre, www.ccdc.cam.ac.uk/structures.

## 4. Conclusions

In summary, we have demonstrated the oxidative reactivity of the chiral N-heterocyclic stannylene, **1**, with the elemental chalcogenide species $O_2$, $S_8$, $Se_\infty$, and $Te_\infty$, respectively. Interestingly, the presence or absence of methyl groups on the $C_2$-bridge backbone of the stannylenes **1** and **2** has a significant effect on the molecular structure of the stannylenes. The preliminary results reported here also hint at the improved stability of 1, which we believe is a direct result of the presence of the methyl groups on the $C_2$-bridge backbone of the stannylenes.

Single crystal X-ray diffraction studies of the Sn(IV)-chalcogenide products from the direct reaction of **1** with $O_2$, $S_8$, $Se_\infty$, and $Te_\infty$, respectively, showed the complexes **3–6** to possess either a trimeric structure, in the case of the oxo-derivative **3**, comprised of a central $\{Sn_3O_3\}$ core, and dimeric structures in the case of the sulphide, selenide, and telluride (**4–6**) comprising a central $\{Sn_2E_2\}$ (E = S, Se or Te).

Work is ongoing toward probing the comparative reactivity of the Sn, Ge, and Pb analogues of N-heterocyclic carbene analogues, with and without methyl groups, on the $C_2$-bridge backbone as well as with differing N-substituents.

**Supplementary Materials:** The following supporting information can be downloaded at: https://www.mdpi.com/article/10.3390/inorganics11080318/s1, **Table S1**: Experimental Single-crystal X-ray crystallographic parameters for complexes **2-6**.

**Author Contributions:** Conceptualization, A.L.J.; methodology, K.R.F. and J.D.P.; Crystallographic analysis, A.L.J., J.D.P. and G.K.-K.; NMR Investigation, K.R.F. and J.D.P.; writing—original draft preparation, A.L.J.; writing—review and editing, A.L.J.; supervision, A.L.J.; project administration, A.L.J.; funding acquisition, A.L.J. All authors have read and agreed to the published version of the manuscript.

**Funding:** This research was funded by the University of Bath, Bath UK with a PhD studentship to K.R.F.

**Data Availability Statement:** The data presented in this study are available on request from the corresponding author.

**Conflicts of Interest:** The authors declare no conflict of interest.

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
