# Peer review of "Reactivity of N-Heterocyclic Stannylenes: Oxidative Addition of Chalcogen Elements to a Chiral NH-Sn System"

_inorganics, doi:10.3390/inorganics11080318_

Round 1

Reviewer 1 Report

This work presents an investigation into the reactivity of a racemic N-heterocyclic stannylene complex with chalcogen elements, resulting in the formation of di- and tri-nuclear compounds. The newly formed complexes have been thoroughly characterized, considering the potential isomers and their corresponding geometries. Based on the findings, I believe the study constitutes a significant contribution to this field of chemistry, and it is likely to be of interest to scholars of the sector.

Therefore, I recommend publishing it with the following minor revisions.

line 81: the experimental section lacks mention of the use of THF, which contradicts the procedure reported here.

lines 90-91: This observation is expected due to the C2 symmetry of the complex, resulting in the presence of four non-equivalent proton couplings.

Scheme 4: The second product after the second arrow cannot be a radical since it violates the requirement for an even total count of valence electrons.

Line 235: The experimental section contradicts the reported procedure as it indicates the use of either 1 equivalent or a slight excess of the reagent.

Scheme 6. There is no need to distinguish between the two species, R2S2 and S2R2, as they are identical and can be superimposed.

Line 392: the yield of complex 6 is significantly low. Is this attributed to poor selectivity of the reaction? Furthermore, do the authors have any insights regarding the presence of other products in the reaction mixture?

Author Response

Please see the attached responce.

Reviewer 2 Report

The manuscript by Johnson and co-workers report the oxidative addition of group 16 elements to an already known N-heterocyclic stannylene that bears tert-butyl substituents on the N-atoms, and methyl substituents on the CH-CH backbone of the five membered ring. In case of molecular oxygen (excess) a trimeric oxo-complex has been isolated whereas reacting two equivalents of S, Se, and Te, the corresponding distannachalcogenides have been isolated. Identical reactions with an N-heterocyclic stannylene lacking methyl substituents on the CH-CH backbone, didn't lead to isolable products. The work has been nicely carried out and definitely deserves to appear in Inorganics after a careful revision.

1. There are numerous grammatical and formatting mistakes which should be avoided in the revision.

2. Please specify x and y on page 2 in [MxEy]n.

3. Please add a reference to justify the claim that, "such systems have potential utility in the formation of Sn(II) chalcogenide thin films and nanocrystal formation" on page 2, line 59.

4. In the caption of scheme 1, please remove R` = Me and R` = H.

5. NMR data have been already incorporated in the experimental section so there is no need to add Table 1. I would recommend to replace it with Table S1 from supporting information.

6. Please provide supporting information as no such file has been provided.

7. Please provide checkcif files for review process.

8. Could the authors please comment on the low yield (9 %) and purity of 6. Could it be that the low yield and low carbon content is due to the formation of other products such as cluster compounds! It might be interesting to run the PXRD and compare it with the simulated one from that of single crystal.

9. In conclusion, I agree that the presence/absence of methyl groups effect the molecular structures of 1 and 2 but the effect on reactivity and stability is somehow premature without additional studies. For instance, what happens if steric bulk is varied on N-atoms in the presence of methyl groups on CH-CH backbone? If it is true then shouldn't the yield of 6 be comparatively higher? 

10. Could the authors please also comment that how selective is the formation of distannachalcogenides! For instance, what happens if more than two equivalents of chalcogens (S, Se or Te) are reacted with 1?

There are numerous grammatical and formatting mistakes which needs attention. Some of these errors/mistakes are pointed out below. Please omit rest in the revised version.

1. Abstract: i) "NMR studies (1H 13C and 119Sn) shows..." should be' "NMR studies (1H, 13C and 119Sn) show...". ii) "diffraction studies has been " should be '"diffraction studies have been ".

2. Commas are missing on numerous occasions. For instance, line 30 [plumbene[18], respectively] and 31 [E = Si, Ge, Sn or Pb].

3. Please avoid frequent use and repetition of identical words. For instance "While" has been used too often throughout the manuscript.

4. Some sentences are too long and need rephrasing. For instance; i) Line 26 to 30, "While........respectively." ii) Line 31 to 34. iii) Line 86 to 91.

5. Spectrum is singular and spectra is plural so please pay attention.

6. Please check carefully formatting mistakes. For instance, C6D6 should be C6D6.

7. Line 47, file should be instead film.

8. Line 38, please delete of from, "of are".

9. Scheme 4, "predicted" is written twice.

Author Response

please see tha attached responces.

ALJ

Round 2

Reviewer 2 Report

Authors have addressed all the necessary changes in the revised version. I am looking forward to see this nice work in Inorganics.